# Group II Oxide Grains: How Massive Are Their AGB Star Progenitors?

**Sara Palmerini** [1,2,*], **Sergio Cristallo** [2,3], **Luciano Piersanti** [2,3], **Diego Vescovi** [2,4] **and Maurizio Busso** [1,2]

1   Dipartimento di Fisica e Geologia, Università Studi di Perugia, Via A. Pascoli snc, 06123 Perugia, Italy; maurizio.busso@pg.infn.it
2   INFN, Sezione di Perugia, Via A. Pascoli snc, 06123 Perugia, Italy; sergio.cristallo@inaf.it (S.C.); luciano.piersanti@inaf.it (L.P.); diego.vescovi@pg.infn.it (D.V.)
3   INAF, Osservatorio d'Abruzzo, Via Mentore Maggini snc, 64100 Teramo, Italy
4   Institute of Applied Physics, Goethe University Frankfurt, Max-von-Laue-Strasse 1, 60438 Frankfurt am Main, Germany
*   Correspondence: sara.palmerini@pg.infn.it

**Abstract:** Presolar grains and their isotopic compositions provide valuable constraints to AGB star nucleosynthesis. However, there is a sample of O- and Al-rich dust, known as group 2 oxide grains, whose origin is difficult to address. On the one hand, the $^{17}O/^{16}O$ isotopic ratios shown by those grains are similar to the ones observed in low-mass red giant stars. On the other hand, their large $^{18}O$ depletion and $^{26}Al$ enrichment are challenging to account for. Two different classes of AGB stars have been proposed as progenitors of this kind of stellar dust: intermediate mass AGBs with hot bottom burning, or low mass AGBs where deep mixing is at play. Our models of low-mass AGB stars with a bottom-up deep mixing are shown to be likely progenitors of group 2 grains, reproducing together the $^{17}O/^{16}O$, $^{18}O/^{16}O$ and $^{26}Al/^{27}Al$ values found in those grains and being less sensitive to nuclear physics inputs than our intermediate-mass models with hot bottom burning.

**Keywords:** AGB star; presolar grain; Nucleosynthesis-Star: abundances; reaction rate; isotopic abundance





## 1. Introduction

The Asymptotic Giant Branch (AGB) is an advanced phase of stellar evolution experienced by low and intermediate mass objects [1–4]. Mid-infrared observations show the presence of alumina dust (amorphous $Al_2O_3$) in the extended envelopes of O-rich AGB stars. Although only recent condensation experiments confirmed that amorphous alumina can condense in circumstellar conditions [5], intermediate mass (IM) AGB stars have been considered to be the main source of alumina dust in the galaxy for many years, starting from the first work by Onaka et al. [6], to the most recent ones Dell'Agli et al. [7], Ventura et al. [8] (and the references therein).

It is, therefore, reasonable to hypothesize that IM AGBs are also the progenitors of the presolar oxide (mainly spinel $MgAl_2O_4$ and corundum $Al_2O_3$) and silicate grains (both amorphous or crystalline) found in pristine meteorites [9]. However, since their first isolation, these presolar grains have been recognized to mainly form in low mass (LM) AGBs ($\lesssim 1.5\,M_\odot$) and, to a lower extent, in IM-AGB stars (4–7 $M_\odot$) [10]. These are the mass ranges within which AGB stars preserve O-rich envelopes until the final formation of a planetary nebula—an environmental condition necessary for the oxide grain condensation.

In principle, the $^{17}O/^{16}O$ ratios together with the $^{26}Mg$ excesses recorded in each grain (hinting to a formation in a $^{26}Al$-rich environment) indicate the initial mass of the progenitor stars. The situation is complicated by the fact that, in order to account for the grain oxygen and aluminum isotopic ratios, H-burning at the base of the envelope must occur coupled to convection in IM-AGBs (hot bottom burning, HBB; [4,11,12]) or to non-convective mixing in LM-AGBs (Cool Bottom Process, CBP; [13]).

In the first case, the efficient convective motions in the envelope prevent stars from becoming C-rich and deplete [17]O and [18]O surface abundances[1]. On the other side, CBP replenishes the cold envelope of LM stars with the fresh ashes of the H-burning shell. Among the issues that make the identification of stellar environments suitable to form grains of group 2, there is the fact that, apart from O, Al, and Mg, no trace of other elements can help in discriminating among the proposed scenarios (as occurs with s-elements in mainstream presolar SiC grains [9]).

The number of presolar oxide grains collected has grown over the years, and today's sample (as reported by the Presolar Grain Database of the Laboratory for Space Sciences of the Washington University in St. Louis [14]) is shown in Figure 1. The classification in four oxide grain groups is reported as well [15]. The few grains studied by Boothroyd et al. [10,11] and Wasserburg et al. [13] have been classified as belonging to group 1, the largest one, whose progenitors are presumably red giant branch (RGB) and asymptotic giant branch (AGB) stars, as the oxygen isotopic ratios observed in these stars [16,17] are very similar to the ones measured in those grains. Alternatively, other sources for a few group 1 grains have been proposed, as supernovae [18] or novae, for dust characterized by an extreme [17]O enrichment [19].

Over the years, silicate grains have also been collected in the presolar grain database of the Washington University in St. Louis [14]. As it is shown in Figure 2A, they have the same oxygen isotopic mix of the oxide grains, and they can be classified in four groups as well. Although, the distribution of the silicate grains in the four groups is not the same of the oxide ones. In particular, group 4 silicates are overabundant compared to those of groups 1 and 2 (likely because of the injection of materials from a nearby supernova into the solar nebula [20]); while group 2 silicates are rarer than oxides. Such a lack could be accounted for by the fact that silicates are expected to form mainly in IM AGBs affected by HBB [7,8].

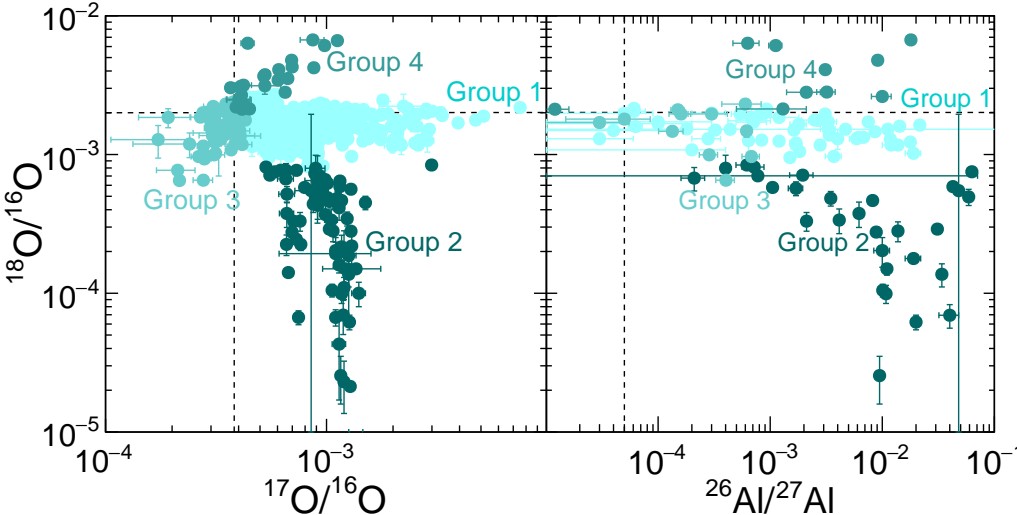

**Figure 1.** Left panel: oxygen isotopic ratios in presolar oxide grains. Labels and colors identify the grains belonging to different groups according to Nittler et al. [15]. The compositions and the stellar origins of group 1 and 2 grains are discussed in detail in the paper. Group 3 grains are supposed to reflect in their isotopic mix the evolution of the oxygen isotope abundances in the Galaxy, forming in low metallicity stars [21], while group 4 grains likely formed in supernovae [22]. Right panel: [18]O/[16]O vs. [26]Al/[27]Al isotopic ratios recorded in grains of the same sample. Different colors identify grains belonging the 4 groups. The black dashed lines indicate the solar values for the O and Al isotopic ratios.

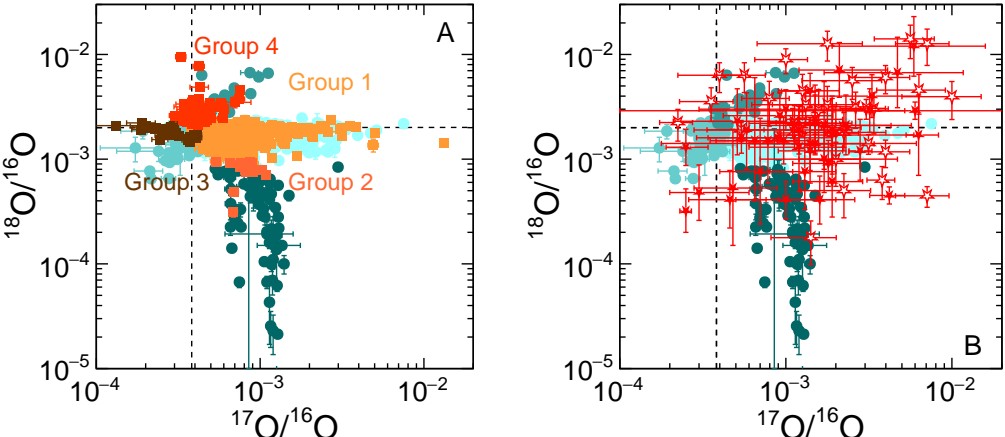

**Figure 2.** (**A**) Oxygen isotopic ratios in presolar silicate grains, from the WUSTL database [14] (orange squared symbols). Labels and colors identify grains belonging to different groups as it is for oxide grains (cyan dots) in Figure 1. Silicate grains overlap to the oxide one in the oxygen three-isotope plot. (**B**) Comparison between the oxygen isotopic ratios of oxide grains (the same of Figure 1) and the abundances measured in AGB stars by (Hinkle et al. [23] open red stars) and (Lebzelter et al. [24] full red stars).

According to the WUSTL Database[2], traces of Mg and Al have been so far recorded only in oxide grains and not in silicate ones. As we will discuss in the next sections (and as illustrated by Figures 5 and 6), the isotopic ratio of aluminum plays a fundamental role in investigating the mass of the stellar progenitors of the grains. Therefore, in this note, we focus on oxide grains only, and in particular those belonging to group 2, which have retained AGB origins and show $^{17}O/^{16}O$ ratios larger than the solar value and $^{18}O/^{16}O$ $\leq 0.001$. Such values, as well as those of $^{26}Al/^{27}Al$ reported in the right panel of Figure 1, are theoretically predicted by AGB models for two ranges of mass: objects with an initial mass $\leq 1.5$ M$_\odot$ affected by CBP (or other forms of non-convective deep mixing) or more massive stars (M $> 4$ M$_\odot$) in which HBB is at play.

After more than 25 years of studies in the field, the two scenarios remain valid. Good fits to the distributions of oxygen isotopic abundance recorded in group 2 grains have been provided by IM-AGBs with HBB [25] as well as by LM-AGBs with CBP ($\leq 1.5$ M$_\odot$) [26]. To distinguish between the two hypotheses remains a difficult topic, given the level of uncertainty in computing stellar models, as well as in grain data measurements (despite their larger accuracy with respect to spectroscopic observations). However, a key role in identifying the masses of stellar progenitors is also played by nuclear physics.

Since the oxygen isotopic ratios in oxide grains largely depend on the reaction rates adopted, a detailed knowledge of nuclear inputs is mandatory. As a matter of fact, in the past, different sets of reaction rates (in particular for proton captures on oxygen isotopes) have been used: Palmerini et al. [26] adopted data based on the Trojan Horse Method [27,28], while Lugaro et al. [25] employed reaction rates measured by direct techniques with the LUNA experiment [29]. In Table 1 we report the two sets: Set A is an updated version of the one used by Palmerini et al. [26], while set B is an updated version of the one used by Lugaro et al. [25].

**Table 1.** Set A and set B of the reaction rates employed in nucleosynthesis calculations.

| Reaction | Set A | Set B |
|----------|-------|-------|
| $^{16}O(p,\gamma)^{17}F$ | Iliadis et al. [30] | Iliadis et al. [30] |
| $^{17}O(p,\alpha)^{14}N$ | Sergi et al. [31] | Bruno et al. [32] |
| $^{17}O(p,\gamma)^{18}F$ | Sergi et al. [31] | Di Leva et al. [33] |
| $^{18}O(p,\alpha)^{15}N$ | La Cognata et al. [34] | Bruno et al. [32] |
| $^{18}O(p,\gamma)^{19}F$ | Iliadis et al. [30] | Best et al. [35] |
| $^{25}Mg(p,\gamma)^{26}Al$ | Straniero et al. [36] | Straniero et al. [36] |
| $^{26}Al(p,\gamma)^{27}Si$ | Iliadis et al. [30] | Iliadis et al. [30] |

Considering the new nuclear data that ahs become available in the last years, we present a comparison between LM-AGBs and IM-AGBs nucleosynthesis computed with both nuclear data sets, in order to determine if such a comparison may help in discriminating between LM-AGBs and IM-AGBs as parent stars of presolar oxide grains of group 2. The LM stellar models used in this paper are the same as in Palmerini et al. [26] (in which the magnetic mixing is modeled by running the MAGIC post-process code [37] on stellar structure computed by the FUNS code [3]) but with a partial update of the employed nuclear physics input. The IM-AGB calculations differ from the ones by Lugaro et al. [25] for both the nuclear physics inputs and the adopted stellar code (we use the FUNS code by Straniero et al. [3]; see also Section 4 for details).

## 2. Oxygen Isotopic Ratios Observed in the Spectra of Present-Day AGB Stars

Before discussing in detail the analysis that leads us to the conclusion that LM-AGBs are the progenitor of group 2 oxide grains, we spend a few words discussing whether the comparison of the presolar grain abundances with astronomical observations of AGB stars could (or not) help us in selecting the mass range of the dust stellar progenitors. Up to date, no measurement of the aluminum isotopic ratio in stellar spectra is available, and oxygen isotopic ratios have been measured only in the spectra of LMS. Observations of oxygen isotopic abundances in dusty and IM ($M \leq 4\,M_\odot$) HBB AGB stars are indeed hampered by difficulties in modeling the atmospheres of these stars, especially in the near-IR where the CNO isotopic ratios can be measured.

Recent studies of variable (Mira, SRa-, and Lb-type) AGB stars [23,24] have found that the $^{17}O/^{16}O$ values recorded for a relatively large ($\sim$77) sample of stars range between $2.5 \times 10^{-4}$ and $6 \times 10^{-3}$ and, therefore, agree with the ones recorded in group 1 oxide grains and the predictions of LM stellar models. In Figure 2B, the oxygen isotopic ratios recorded in oxide grains are compared with the abundances in the AGB spectra (by [23,24]). The values of the $^{17}O/^{16}O$ isotopic ratio measured in the stars covers the whole range of group 1 oxide grains. The $^{18}O/^{16}O$ isotopic ratio of part of the stellar sample is similar to that of group 1 grains; however, the rest of the stars show instead $^{18}O/^{16}O$ ratios larger than the solar values (and thus of the grains).

The grains with $^{18}O/^{16}O \leq 10^{-4}$ are not reached by the stars. A possible shift of the stellar $^{18}O/^{16}O$ abundances with respect to the grains could be accounted for by galactic chemical evolution. Indeed, the initial composition of the observed stars (which are present day AGBs formed later than the birth of the solar system) should, in principle, be different from that of the grain progenitors that ended their life a few $10^8$ yr before the formation of the Sun [23]. Only a few stars in Figure 2B overlap to group 2 grains, and this observational fact might cast doubts on the possibility that LM AGB stars are the progenitor of this type of dust. Hinkle et al. [23] suggested that this is the observational evidence that the extra-mixing process is hampered at high metallicity.

Nevertheless, most of the objects in the sample are M stars with mass $\leq 2.5\,M_\odot$ at the beginning of AGB evolution that have not yet experienced TDU. Thus, also extra-mixing has not yet had much time to occur during the AGB phase and the stellar surface abundances are almost the same as those at the end of the RGB phase. In star climbing the

red giant branch for the first time, the extra-mixing is much less efficient in destroying $^{18}O$ because the temperature of the H-burning shell is lower than during the AGB phase.

The authors in Hinkle et al. [23] and Lebzelter et al. [24] stated that no stars affected by HBB were observed. Moreover, the most massive objects in their samples (which also include a few S-stars) may have not experienced CBP during the RGB phase. Indeed, the larger the stellar mass is, the shorter the time in which extra-mixing can work. This is due to the fact that the discontinuity left by the FDU is reached and removed by the H-burning shell later in this phase, with the luminosity bump not occurring at all in stars more massive than $\sim 2.2$ M$_\odot$. Such a limit is even smaller in object of high metallicity ([37] and the references therein).

Finally, a consideration has to be made on the error bars, which are shown in Figure 2 for both the meteoritic and astronomical data. The uncertainties of the latter are (in most cases) much larger than those of the former. Therefore, although the observations of oxygen isotopic ratios in LM AGBs are valuable data, their precision is not good enough to unambiguously confirm (or reject) that their oxygen isotope mix is compatible with that of group 2 oxide grains.

In particular, the available astronomical data appear to suggest that LM AGBs do not explain the most extreme and $^{18}O$-depleted group 2 grains with $^{18}O/^{16}O \leq 10^{-4}$. However, one might speculate that this value is the lower limit within which it is currently possible to measure the $^{18}O/^{16}O$ ratio in the spectrum of AGB stars. We remember that in Table 5 of Lebzelter et al. [24] the authors discussed how the values of oxygen isotopic ratios, and related uncertainties, can be very different for the same star, depending on the technique (curve of growth or spectrum synthesis) employed in chemical abundance analysis.

### 3. $^{17}O$ + p Reaction Rates and the $^{17}O/^{16}O$ Ratio as Probe of the H-Burning Temperature

Isotopic oxygen and aluminum abundances similar to the ones reported in Figure 1 are found in hydrogen burning regions (e.g., the H-shell of AGB stars), where the temperature exceeds some $10^7$ K. The higher the temperature is, the smaller the equilibrium value of the $^{18}O/^{16}O$ ratio, and the larger the value of the $^{26}Al/^{27}Al$ ratio. The dependence of the $^{17}O/^{16}O$ isotopic ratio on the temperature is more complex (see Figure 3). Since, at low energies, $^{16}O$ is not efficiently destroyed via the $^{16}O(p,\gamma)^{17}F$ reaction [38], it is the proton capture rate on $^{17}O$ that establishes the equilibrium value for the $^{17}O/^{16}O$ ratio at a given temperature. Indeed, such a number can be easily computed as $\frac{^{17}O}{^{16}O} = \frac{R[^{16}O(p,\gamma)^{17}F]}{R[^{17}O(p,\gamma)^{18}F]+R[^{17}O(p,\alpha)^{14}N]}$.

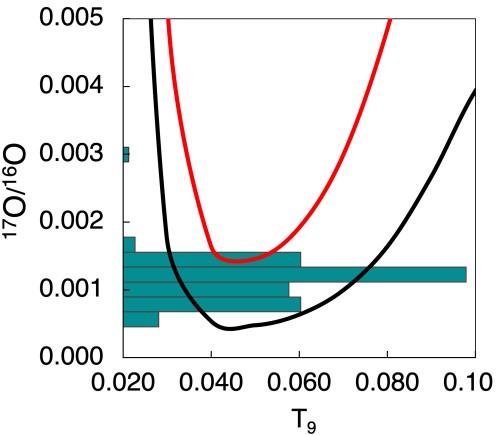

**Figure 3.** The $^{17}O/^{16}O$ equilibrium values as a function of temperature (in units of $10^9$ K) computed using the reaction rates of set A (red curve) and set B (black curve). We report the calculation results only for the rate recommended values, being these former the inputs employed for nucleosynthesis calculation. The cyan histogram marks the $^{17}O/^{16}O$ distribution in group 2 oxide grains.

A high precision estimate of the $^{17}O$ + p rate is crucial to properly use the $^{17}O/^{16}O$ abundance as a thermometer of the stellar environment. From an inspection of Figure 3, it

is evident that with the reaction rates of set A, one reproduces only the higher part of the measured distribution, while, with those of set B, the lower part is preferentially covered. Furthermore, set B allows to cover almost the entire range of $^{17}O/^{16}O$ values recorded in the grains, albeit at different temperatures.

## 4. Deep Mixing and the Formation of Oxide Grains in Low Mass AGB Stars

The First Dredge-up (FDU)[3] sets the initial abundance of oxygen isotopes in the envelope of red giant stars. Apart from an obvious dilution factor, the $^{17}O/^{16}O$ value is determined by the H-burning temperature, and hence by the initial stellar mass. This was extensively discussed by Abia et al. [17], which precisely estimated the stellar mass of $\alpha$-Bootis and $\alpha$-Tauri (with an accuracy of 0.1 M$_\odot$), starting form the oxygen isotopic abundance observed in these red giants (see also [39]).

According to standard stellar models, the oxygen isotopic mix in the envelopes of LM stars ascending the giant branch (both for the first and for the second time) remains unchanged after the dredge up, unless CBP or HBB takes place.

Boothroyd et al. [10] and Wasserburg et al. [13] suggested that stars with mass $\leq 2$ M$_\odot$ accounted only for $^{17}O/^{16}O$ ratios between $10^{-3}$ and $2.2 \times 10^{-3}$, while more massive red giants were the progenitors of grains showing larger values of $^{17}O/^{16}O$. In this scenario, LM RGB stars would be possible progenitors of just a small portion of group 1 grains. The situation changed thanks to a new measurement of the $^{14}N(p,\gamma)^{15}O$ reaction rate [40], which turned out to be 50% smaller and, as a consequence, would force stars to burn hydrogen at a higher density and temperature than previously envisaged (25% and the 10%, respectively; see [37]).

Once computed with this new rate, the $^{17}O/^{16}O$ abundance ratio left by the FDU in the envelope of a 2 M$_\odot$ star with solar metallicity moves from $2.2 \times 10^{-3}$ to $5.14 \times 10^{-3}$, covering the whole range of the $^{17}O/^{16}O$ values measured in presolar oxide grains of group 1. If deep mixing processes start with these initial conditions, it is found that a rather shallow CBP at play during the RGB succeeds in covering also the sub-solar values of the $^{18}O/^{16}O$ isotopic ratio measured in group 1 oxide grains (see the gray curves in Figure 4, showing the lower limit of the $^{18}O/^{16}O$ isotopic ratios that can be accounted for CBP during the RGB phase).

Therefore, RGB stars with mass $\leq 2$ M$_\odot$ turn out to be reliable progenitors of the majority of group 1 oxide grains [37]. In group 2 oxide grains, the $^{17}O/^{16}O$ isotopic ratio ranges from $5 \times 10^{-4}$ to $2 \times 10^{-3}$ (apart from one single grain with $^{17}O/^{16}O = 3 \times 10^{-3}$). Conversely, the measured values of $^{18}O/^{16}O$ and $^{26}Al/^{27}Al$ are more puzzling. As we need to achieve $^{26}Al/^{27}Al$ ratios $\geq 3 \times 10^{-3}$ and $^{18}O/^{16}O \leq 10^{-4}$, the H-burning has to occur at $T \geq 4 \times 10^7$ K. How can materials so rich in $^{26}Al$ be present in the envelopes of evolved stars with masses smaller than 2 M$_\odot$?

Following the suggestion by Wasserburg et al. [13], Nollett et al. [41] demonstrated that the oxide grain isotopic ratios could be accounted for by the effects of CBP on the AGB envelope abundances. Indeed the conveyor belt mixing model (developed by those authors) could efficiently synthesize $^{26}Al$, deplete $^{18}O$ and produce or destroy $^{17}O$ according to the mixing depth. However, two problems do not allow the agreement to be completely satisfactory:

- In the $^{18}O/^{16}O$ vs. $^{17}O/^{16}O$ plane, several grains occupy a *forbidden* area (at $^{17}O/^{16}O < 0.0005$ and $^{18}O/^{16}O < 0.0015$) that is not accessible by CBP nor by HBB models.
- To account for the highest $^{26}Al/^{27}Al$ values found in oxide grains, the CBP has to reach the most energetic layers of the H-burning shell, but this would imply an appreciable feedback on the stellar energy balance.

The first issue was solved by updating the proton capture cross sections on $^{17}O$ and $^{18}O$ [27,34]. Therefore, it is possible to state that group 1 dust particles might form in RGB stars with a mass smaller than 2 M$_\odot$ and that group 2 grains form in AGB stars of $1.2-1.5$ M$_\odot$ both experiencing CBP [27].

The problem of accessing $^{26}$Al/$^{27}$Al values larger than a few $10^{-3}$ remains instead a severe limit for CBP. This is due to the physics of the mixing itself. Indeed, classic CBP is a non-convective mixing phenomenon operated by conveyor belts, which bring materials from the cool bottom of the convective envelope down to the inner stellar layer near the hydrogen burning shell, where they collect the nucleosynthesis ashes, dragging them upwards and enriching the stellar envelope, which assumes a composition gradually approaching the one of H-burning regions [13,27,37,41]. Mixing of material very close to the H-shell would lead to a feedback on the surface luminosity that must be properly taken into account in the stellar model.

Deep mixing phenomena and their possible physical causes have been investigated by several authors [37,41–45]. Among the suggested mechanisms, it was noted that some form of *themorhaline diffusion* is induced in stars by the burning of $^3$He via the $^3$He($^3$He,2p)$^4$He reaction, which leads to a local inversion of the average molecular weight in a peripheral layer of the H-burning regions. This determines the descent of the overlying material (on average density) triggering a diffusive mixing between the base of the convective envelope and the H-shell [45].

However, thermohaline diffusion was demonstrated to be insufficient to account for the oxide grain composition because the $^3$He($^3$He,2p)$^4$He occurs at temperatures $\leq 3 - 3.5 \times 10^7$ K, while temperatures $\geq 4 - 5 \times 10^7$ K are required to reproduce the oxygen and aluminum isotopic ratios recorded in group 2 grains [42,46]. Another problem of this process is that it occurs on very long time scales, with extremely small velocities, which are likely insufficient to induce large changes in evolved evolutionary phases [47].

At the moment at which this note was prepared, the only LM-AGB nucleosynthesis model able to simultaneously reproduce the $^{17}$O/$^{16}$O, the $^{18}$O/$^{16}$O, and the $^{26}$Al/$^{27}$Al ratios measured in group 2 oxide grains was the one suggested by Busso et al. [48] and formalized by Nucci and Busso [49]. In this scenario, the natural expansion of magnetized structures from above the H-burning shell induces a bottom-up mixing, which pushes into the stellar envelope materials so enriched in $^{26}$Al to reproduce the values recorded in the extremely $^{26}$Al-rich grains (up to $^{26}$Al/$^{27}$Al of a few $10^{-2}$; see [26]). This is due to the fact that advection of magnetic bubbles into the envelope powers a mixing different from the classical forms of deep-mixing, based on the downward penetration of heavy materials.

In this case, instead, the diffusive down flow of material is triggered by the relatively fast up-flow of the magnetic bubbles [50,51]. In this way, the "hot" rising matter has no time to burn along the path and, in its rise, pushes down unprocessed materials from the envelope. As we will discuss in detail in Section 5, the highest $^{26}$Al/$^{27}$Al value reached by our HBB models is 0.1, while the highest value reported in Figure 4 of Lugaro et al. [25] is 0.07.

Although these values are sufficient to reproduce a large part of the $^{26}$Al/$^{27}$Al spread shown by group 2 oxide grains, both our models and those of Lugaro et al. [25] reach the highest $^{26}$Al abundances when they are already depleted in $^{18}$O and, thus, unable to match the grains. To solve this problem, it is, therefore, necessary to introduce dilution effects and/or to suppose that a contamination during the measurement has led to an overestimation (by some orders of magnitude) of the $^{18}$O/$^{16}$O ratios.

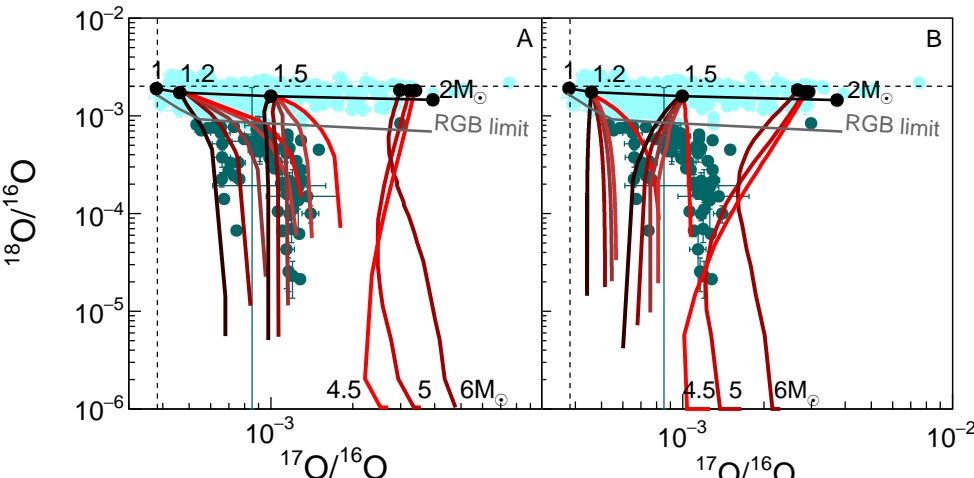

**Figure 4.** The $^{18}O/^{16}O$ vs. $^{17}O/^{16}O$ isotopic ratios in the envelope of solar metallicity stars at FDU (black solid line and markers) with mass from 1 to 2 $M_\odot$ and at SDU for masses from 4.5 to 6 $M_\odot$, as indicated by the labels. Light cyan points are group 1 oxide grains, while darker dots are those of group 2. The red curves descending from the 1.2 $M_\odot$, and the 1.5 $M_\odot$ markers refer to magnetic mixing model results for AGB stars for different values of $k$ (see the text for details). The maximum modification of the envelope composition that can be produced by CBP during the RGB phase is indicated by the gray curve. Red curves starting from 4.5, 5, and 6 $M_\odot$ SDU abundances deal with the evolution of $^{17}O/^{16}O$ and $^{18}O/^{16}O$ isotopic ratios in the envelope of AGB stars affected by HBB. Panel (**A**) refers to calculations run with reaction rates of set A and Panel (**B**) to calculations run with the set B. The dashed lines mark the $^{17}O/^{16}O$ and the $^{18}O/^{16}O$ solar values.

In running our calculations, we apply the magnetic extra-mixing model to 1.2 and 1.5 $M_\odot$ AGB stars with solar metallicity. The mixing velocity is determined by the velocity by which magnetized bubbles cross the region between the H-burning shell and the base of the convective envelope, namely $v(r) = v(r_k)(\frac{r_k}{r})^{k+1}$ [49], where $r$ is the position along the stellar radius and $k$ is the index of the power law $\rho \propto r^k$, which deals with the density distributions in the crossed region. When the H-shell is burning, $k$ remains constantly $\sim-3$ from the layer immediately below the envelope down to the deeper layers just above the burning shell.

Thus, $k$ can be use to select the deepest layer from which the mixing starts and $r_k$ and $v(r_k)$ are the mixing starting depth and starting velocity (see [26] for more details). Moreover, assuming that the bubble will cross the bottom of the envelope with a velocity a bit smaller than the velocity of the deepest convective layer (e.g., $v(r_e) \sim 10^4$ cm s$^{-1}$), we obtain $v(r_k)$ of the order of a few tenths cm s$^{-1}$. To complete the model, the mixing rate is estimated as $\dot{M} = 4\pi\rho_e^2 v(r_e)f_1f_2$, being $\rho_e$ is the density of the radiative layers just below the border of the convective envelope, while $f_1 \sim (0.01 \div 0.02)$ and $f_2 \sim 0.01$ are the fraction of the stellar surface covered by the magnetized bubbles and their filling factor, respectively [50].

In Figure 4 we report the evolution of the oxygen isotopic ratios in the stellar envelopes of our AGB star models. Panel A and B show nucleosynthesis calculations run with nuclear reaction rates from set A and B, respectively. The descending red curves show the evolution of the oxygen isotopic mix due to magnetic mixing during the AGB phase, starting from the surface composition left by the FDU and adopting different values of $k$ ($-3.5$, $-3.4$, $-3.3$, $-3.2$, and $-3.1$, with $-3.1$ as the largest $k$ value in the radiative layers below the convective envelope of the stellar models employed in the calculations—see [26] for details). The smaller the value of $k$, the deeper the mixing, the darker the curve.

In the same way, Figures 5 and 6 show the evolution of the surface abundance of $^{18}O/^{16}O$ and $^{17}O/^{16}O$ (respectively) as a function of the $^{26}Al/^{27}Al$ ratio for the same

models. Oxide grains of group 1 (light cyan dots) and group 2 (dark cyan dots) are reported in Figures 4–6 for comparison with model predictions of both LM AGBs (with extra-mixing) and IM AGB (affected by HBB). In LM stellar models, the $k$ values from $-3.5$ to $-3.1$ correspond to small variations of the mixing depth and define a zone thick enough to host different abundances and temperatures.

For this reason, models with a higher $k$ are less efficient in destroying $^{18}O$ and increasing $^{26}Al$. The dependence of the final values of the $^{17}O/^{16}O$ isotopic ratio to the adopted $k$ is instead somewhat more complicated due to the trend of the equilibrium value of this isotopic ratio as a function of the temperature, which is not linear (as discussed in Section 3).

From panel A of Figure 4, one can observe that magnetic mixing calculations for a 1.2 $M_\odot$ AGB run with reaction rates of set A account well for the majority of the grains, with small changes in the value of the parameter $k$. On the contrary, in panel B, the 1.2 $M_\odot$ AGB run with reaction rates of set B only matches a small portion of the oxygen isotopic ratios of group 2 grains. Thus, the contribution of the 1.5 $M_\odot$ stars (and maybe also that of a more massive object) is needed to cover the portion of the plot occupied by the grains.

The main difference between the nucleosynthesis calculations shown in the two panels of Figure 4 is mainly ascribed to the adopted $^{17}O(p,\alpha)^{14}N$ reaction rate. The $^{18}O+ p$ rates of set A, which are largely less efficient than the ones of set B, just lead to a smaller depletion of the $^{18}O/^{16}O$ ratio in the stellar envelope (which is, in any case, very low).

Independently from the chosen reaction rate, both the 1.2 $M_\odot$ AGB (with set A) and the 1.5 $M_\odot$ (with set B) seem valuable candidates to be progenitors of group 2 grains, because the whole range of the $^{26}Al/^{27}Al$ isotopic ratio recorded in the grains is covered by both models (see Figures 5 and 6). However, in the figures, we only report the models that better match the oxygen isotopic mix of the grains in Figure 4. Therefore, even if the agreement between predictions and observations in Figure 5A is slightly worse than in Figure 5B, the 1.2 $M_\odot$ model run with set A turned out to be our best candidate as a progenitor for group 2 oxide grains (since it provides the best fit to the oxygen isotopic ratios).

In the present analysis, the role of neutron captures on $^{26}Al$ has not been discussed because the LM stars we considered are characterized by a very low number of TDU episodes. As a consequence, the effects induced by neutron capture nucleosynthesis is very small in the envelope, and therefore the contribution of $(n,\gamma)$ reactions on $^{26}Al$ abundances in oxide presolar grains is expected to be almost negligible.

## 5. Hot Bottom Burning in Intermediate Mass Stars

A possible alternative source of presolar group 2 oxide grains is represented by IM-AGB stars [25,52–55]. In fact, in these stars, the temperature at the base of the convective envelope becomes high enough for p-capture reactions to occur efficiently. Such an occurrence, usually addressed as HBB, significantly alters the light isotopes abundances, by producing $^{13}C$, $^{14}N$, $^{26}Al$ and destroying $^{12}C$, $^{18}O$. However, it has to be remarked that, in massive AGBs, in particular with initial metallicity lower than solar, the recurrent occurrence of TDUs increases the carbon budget of the envelope, so that the exact isotopic distribution on the surface does depend on the interplay of HBB and TDU efficiency. As a consequence, depending on which is the dominant mechanism, different types of presolar grains may be produced (with O-rich grains being the dominant population).

Models of IM-AGB stars, however, still suffer of many limitations related to an incomplete knowledge of the physical processes at work. For instance, the mass loss efficiency has a direct impact on the produced nucleosynthesis, because the larger the mass loss, the lower the growth in mass of the degenerate core, the lower the temperature at the base of the convective envelope, and hence the lower the HBB efficiency [56]. On the other hand, the predictive power of the current models of IM-AGB is hampered by several oversimplified physical assumptions adopted in the computations, such as the adopted mixing scheme and its coupling to other physical processes.

In computing these models, it is fundamental to monitor the relative timescale of the p-capture processes with respect to the characterization of the convective mixing. As the

temperature at the base of the convective envelope is very large, the local nuclear timescale becomes comparable with the local convective mixing timescale, so that the equations describing the evolution of matter via nuclear burning and those describing the mixing should be solved simultaneously [55]. In addition, the energy feedback related to the HBB could significantly alter the physical conditions at the base of the convective envelope, so that all the equations describing the physical and chemical structure of a star should be solved simultaneously [57].

By keeping in mind all the previous considerations, we computed few AGB models of stars with the masses 4.5, 5.0, and 6.0 $M_\odot$. We make use of an improved version of the FUNS code [3], previously used to compute FRUITY models [58–62]. With respect to FRUITY, the models presented here adopt the same upgraded physical inputs as in Vescovi et al. [51]. In particular, the use of an improved version of the Equation of State leads to larger temperatures at the base of the convective envelope and, thus, to a more efficient activation of the HBB [57]. A detailed comparison to models by other groups (e.g., [55,63,64]) is beyond the scope of this paper (but see [60]).

We warn the reader that any result, better or worse, related to IM-AGBs has to be taken with a grain of salt, in consideration of the many uncertainties affecting the evolution of these more massive AGBs. In the computation, we adopted an extremely reduced mass-loss rate, in order to increase the number of Thermal Pulses and maximize the effects of HBB. In Figures 4–6, the abundance predictions from these models are compared with those derived from grains.

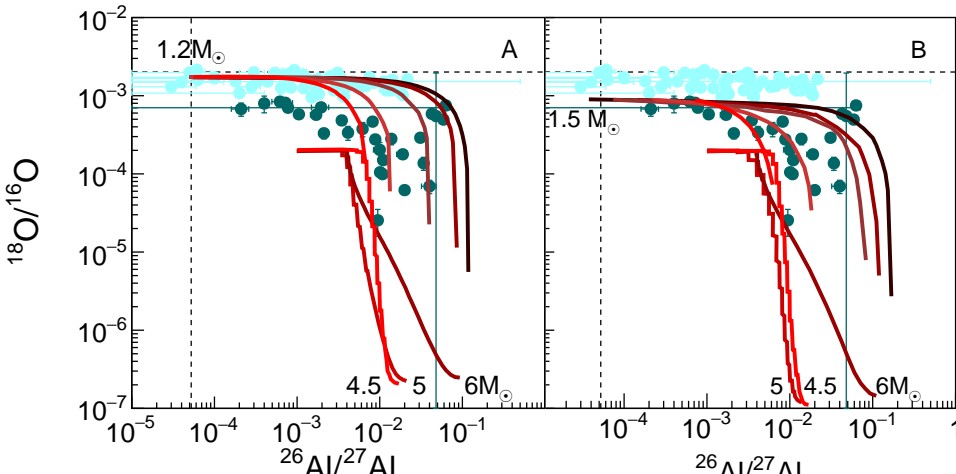

**Figure 5.** The $^{18}O/^{16}O$ isotopic ratio as a function of the $^{26}Al/^{27}Al$ one for the same models and grain data as in Figure 3. Horizontal and vertical dashed lines represent the solar values for the $^{18}O/^{16}O$ and the $^{26}Al/^{27}Al$ isotopic ratios. Panel (**A**) refers to calculations run with reaction rates of set A and Panel (**B**) to calculations run with the set B.

At first glance, these plots could suggest that theoretical models of IM AGBs are unable to reproduce the oxygen ratios measured in oxide grains. While a partial agreement is reached for the $^{17}O/^{16}O$ ratio with 4.5 and 5.0 $M_\odot$ models adopting set B, the final theoretical $^{18}O/^{16}O$ ratios are extremely low ($<10^{-7}$). Interestingly enough, [25] suggested the possibility that material coming from ancient AGB stars may have been diluted with solar-system material. If this is the case, the abundance of isotopes mainly destroyed by HBB, as $^{18}O$, may result as largely increased, while that of isotopes mainly produced in AGBs, as $^{17}O$ and $^{26}Al$, should be only marginally altered. To investigate such a possibility, in Figure 6, we plot the $^{17}O/^{16}O$ ratio versus the $^{26}Al/^{27}Al$ ratio. We focus our attention on the final part of each evolutionary track because stars lose the large majority of their mass during the end of the AGB phase.

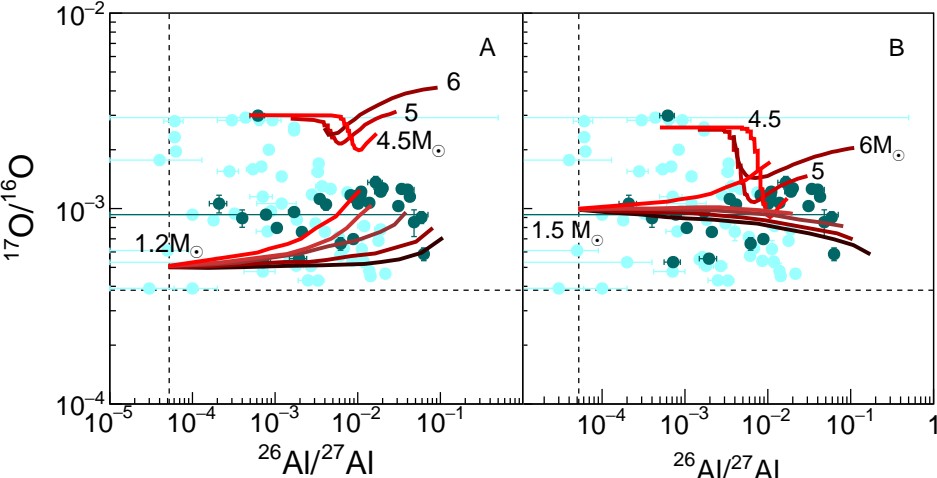

**Figure 6.** The same as Figure 4 but for the $^{17}O/^{16}O$ isotopic ratio versus the $^{26}Al/^{27}Al$ one. Dashed lines mark to the solar values for $^{17}O/^{16}O$ and $^{26}Al/^{27}Al$. Panel (**A**) refers to calculations run with reaction rates of set A and Panel (**B**) to calculations run with the set B.

Figure 6 discloses that, at odds with the 1.2 $M_\odot$ model, the IM-AGB models computed with the set A nuclear cross sections lay outside the region identified by the grains. On the other hand, the IM-AGB models computed with the set B nuclear cross sections partially were able to reproduce the $^{17}O/^{16}O$ ratio observed in grains by assuming a factor 2 uncertainty related to a 50% dilution with solar-system material (see [25] for details). We highlight that our $^{17}O/^{16}O$ ratios are slightly larger than those shown by [25].

This can be ascribed to the different handling of the radiative/convective interface at the inner border of the convective envelope in the framework of the adopted stellar codes. We remark that, in any case, a large portion of grains, i.e. those with the lowest original $^{26}Al/^{27}Al$ ratio cannot be matched by theoretical models, even by assuming the aforementioned dilution factor. The adoption of a stronger mass-loss (e.g., [65]) would not affect our conclusions, because the grain range span from the models would result as decreased.

## 6. Conclusions

Low mass AGB stars with bottom-up deep mixing (similar to the magnetically-induced one) are possible candidates to be progenitors of presolar oxide grains of group 2. This conclusion emerges from the comparison between nucleosynthesis predictions and the isotopic mix recorded in the grains. Indeed, while it is true that the agreement between low mass star models and observations becomes better or worse depending on the reaction rates used, this scenario provides a match to the majority of the grains. A small change, the 25%, in the mass of the progenitor (which, in any case, remains in the range of the LMS) is sufficient to achieve a good fit to the grain abundances.

To enforce this conclusion, other analysis should be performed by adopting different stellar models (e.g., MONSTAR [66], ATON [67], and NuGrid [68,69]). This will help to disentangle the effects due to the nuclear physics input from those related to stellar modeling. In any case, since the abundances of $^{17}O$ and $^{26}Al$ are thermometers of the environments in which these nuclei are synthesized, we are confident that different LM AGB models with bottom-up advective mixing at play (even when not induced by stellar magnetic fields), employing the same nuclear inputs, can provide other accurate fits to group 2 oxide grain compositions.

On the other hand, intermediate FUNS mass AGB models where HBB is at play can reproduce a fraction of the grain samples for one of the two nuclear data sets, while they largely disagree for the other measurements. Once again, we stress that our conclusions

cannot be definitive, since the modeling of IM-AGBs are still affected by large uncertainties related to the physical processes governing their evolution, such as mixing and mass-loss.

**Author Contributions:** Conceptualization and investigation, S.P., S.C., L.P., D.V. and M.B.; software S.P., S.C., L.P. and D.V.; formal analysis, S.P. and S.C.; writing—original draft preparation, S.P. and S.C.; writing—review and editing, S.P., S.C., L.P., D.V. and M.B. All authors have read and agreed to the published version of the manuscript.

**Funding:** D.V. acknowledges the financial support of the German-Israeli Foundation (GIF No. I-1500-303.7/2019).

**Institutional Review Board Statement:** Not applicable.

**Informed Consent Statement:** Not applicable.

**Data Availability Statement:** Presolar grain data are from the Presolar Grain Database of the Laboratory for Space at Wash U Physics http://presolar.wustl.edu (accessed on 11 May 2021). Stellar data are from [23,24].

**Conflicts of Interest:** The authors declare no conflict of interest.

## Notes

1. Note that no extra-mixing mechanism is applied to IM-AGBs.
2. "http://presolar.wustl.edu" (accessed on 11 May 2020) and Hynes and Gyngard [14].
3. A dredge-up episode occurs whenever, following a temporary exhaustion of nuclear burning, the convection extends down from the stellar envelop to the internal layers that have been affected by the burning. As a result, the nucleosynthesis products are mixed into the envelope whose composition is, therefore, modified. The FDU occurs just before a star starts to climb the Red Giant Branch, and the Second Dredge-up (SDU) takes place in objects more massive than $\sim 4\,M_\odot$ at the very beginning of the AGB phase, while the Third Dredge-up (TDU) occurs many times during the AGB phase at the exhaustion of each thermal pulse.

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
