# Peer review of "Group II Oxide Grains: How Massive Are Their AGB Star Progenitors?"

_universe, doi:10.3390/universe7060175_

Round 1

Reviewer 1 Report

My only comment concerns the choice / wording of the title.

Please check if the sense of the title are the two questions:

1) how massive are AGB stars ?   "...how much massive" is no good English.

2) and where do AGB stars come from )  Again, "..AGB stars they come from"

    is no good English.

My suggestion for a slightly revised title:

Oxide grains: how massive are AGB stars and where do they come from?

Author Response

Dear referees,
we are replaying  to all of you with the same letter. We believe this is the best way to proceed giving you a complete explanation of all the changes that you will find in the manuscript.
Indeed you are in 4 and in your reports you have proposed many different corrections and suggestions. Firstly we would like to thank you for the work done, which certainly contributes to improve the quality of the paper.

As usual we have highlighted in bold the corrections along the text. Without listing them one by one, we have reported the minor corrections you suggested and changed the title that is now "Oxide grains: how massive are AGB stars and where do they come from?".

By reading one of the reports it appears that it was not clear the concept that CBP is active only in LM stars while in IM stars only the effect of HBB is considered. It follows that the choice (and discussion) of the values of the parameter k concerns only the models of 1.2 and 1.5Mo and not those of 4.5, 5 and 6Mo. We have tried to stress the concept in the text. In any case, a paragraph is devoted to LMS affected by extra mixing and another one is dedicated to IM AGB with HBB.

Concerning the figures, they have been improved by drawing the error bars of the grain data to highlight that their uncertainties are usually small. Furthermore, a new figure has been added (now figure 2) illustrating in the panel A the presolar silicate grains, which have been also discussed in the introduction as required by referee n.2, while panel B (of Figure 2) shows the comparison between the oxide grains and the observations of oxygen isotopes in M ​​and S stars, as suggested by referee n.3. Moreover Figure 3 was redraw in logarithmic scale.
Finally, we have added the curves dealign with the 1.5Mo in Figure 3A, they were originally not shown because  the 1.2 Mo star fit the data already.

About the silicate grains we specify that they were omitted in the first draft of the manuscript simply because in them  the  26Al / 27Al ratio has not been measured and it provides determined indications on the mass of the stellar progenitor. For this reason we limited the discussion to oxide grains.

A new paragraph has been written on the comparison of grains abundances with astronomical observations of oxygen isotopic ratios in AGB stars. The abundances observed in the AGB samples of [Hinkle et al. 2016, ApJ, 825, 38] and [Lebzelter et al. 2019, ApJ, 886, 117] have been compared with the ones recorded in oxide grains.  A few possible hypotheses on why only part of the stellar sample overlaps with the grains have been discussed. In any case we do not believe that on the basis of these observations it can be stated that low mass AGBs are not the progenitors of group 2 oxide grains.

Finally, we tried to address the request of giving more informations and extending the discussion about the IM modelling and HBB calculations (in the devoted section). Details about the FRUITY code have been added. In the same way  along the text and in the conclusions  we have underline the need of extending and repeating the LM-AGB analysis by employing other stellar codes among those available in literature.

Reviewer 2 Report

This is a very interesting paper that revisits the origin of “Group 2” presolar oxide grains that have large depletions in 18O relative to other presolar oxides and large inferred enrichments in 26Al. Calculations of low-mass AGB stars using the magnetic extra-mixing model provide good fits to the O isotopic compositions and 26Al/27Al ratios of group 2 presolar oxide grains. Additionally, intermediate-mass AGB stars undergoing HBB could also contribute these group 2 grains.

Below are suggested edits for the manuscript.

---------------------------------------------------------------

Line 1 – “their isotopic compositions

Line 2-3 – there are also presolar silicate grains that belong to the group 2 classification. Some also show high 26Al/27Al (e.g., Nguyen and Zinner 2004 Science)

Line 22-24 – as written it is a bit misleading and sounds like more grains derive from IM AGB stars, when low-mass AGB stars actually produce most of the dust.

Line 36-37 – please define what is meant by “exotic elements”, and specify whether you mean elemental or isotopic abundances

Figure 1 – Group 3 grains could either come from low metallicity red giants OR supernova. Please correct this.

Line 38 – “Mainstream”

Line 42 – delete “database” and “typical”

Line 45 – not accurate as written. Group 1 grains are proposed to mainly have RGB and AGB stellar sources, but most recently some have been found to have SN sources (see Leitner and Hoppe, Nature 2019). Also grains with extreme 17O enrichments could be nova grains (e.g., Gyngard et al. POS 2010)

Line 85 – ratios, not abundances.

Line 91 – typo “from” not “form”

Line 103 – again, it should be 17O/16O ratio, not abundance

Lines 105-110 – You mention deep mixing processes and sub-solar 18O/16O ratios, which could also refer to Group 2 grains. Please specify what “deep mixing” refers to here, CBP or SDU/TDU? Perhaps it would be worth stating that CBP also happens on the RGB. Up until now only CBP during the AGB phase has been mentioned, not the RGB. Also specify what is meant by “whole range of values measured”. Do you mean the 17O/16O ratios, 18O/16O ratios, or both?

Are you saying the sub-solar 18O/16O values of group 1 grains can be explained only by shallow CBP? What about the range of initial O compositions of the parent stars and proton capture on 18O and FDU?

Lines 128-129 – previously (line 106) it was stated that shallow CBP in RGB stars was required to explain the sub-solar 18O/16O ratios of group 1 grains. So with the updated proton capture cross sections, is CBP in RGB stars no longer necessary to reproduce group 1 grain compositions?

Figure 3 – it would be helpful to switch the axes, plotting 17O/16O vs. 18O/16O. this is how the O isotopic compositions of presolar silicates and oxides are typically plotted, and how the data are plotted in Lugaro et al., 2017. Also, are all group 2 grains shown here? It might be easier to see using a log scale.

Line 166 – “AGB star” or “to 1.2 and 1.5 solar mass AGB stars”

Line 186 – In the sentence “The smaller the value of k, the darker the curve”, I assume this means -3.5 is the darkest curve, i.e., you are not referring to the absolute value? You may clarify that higher values of k refer to more shallow mixing depths.

Line 199-201 – would the mixing calculation for a 1.2 solar mass star in panel B match the grain data if a broader range of k values were used? But then would these k values be unrealistic in reality? In other words, is k = -3.1 the largest value that can be used because -3 corresponds to the bottom of the envelope?

Was a 1.5 solar mass AGB star not calculated using reaction rates of set A simply because the 1.2 solar mass star fit the data already? It would be useful to illustrate the calculation for a 1.5 solar mass star using set A

Line 200 – Suggest changing to “…of set B only match a portion of the oxygen isotopic ratios of group 2 grains.”

Line 233 – “still suffer from many…”

Line 256 – it is a bit difficult to see the extent of the HBB models in figure 3 because it is on a linear scale. In Fig. 3B, it looks like the 4.5 and 5 solar mass models can reach the compositions of some Group 2 grains. It would be worth discussing this.

For the calculations you chose extremely reduced mass-loss rates to maximize the effects of HBB. So what do the results look like if larger mass-loss rates are chosen? Do you get a better fit to the grain data?

Lines 260-261 – reword for clarity. As written it sounds like the abundance of 18O, which is destroyed by HBB, would be further reduced by the dilution with solar system material, when it is the opposite.

Line 263 – change to “26Al/27Al ratio”

Line 267-268 – the models shown in Figure 3 in Lugaro et al. (2017) extend to lower 17O/16O ratios compared to your calculations. Can you discuss this? Is it because the number of thermal pulses was different in the two studies?

Conclusions – Since set B gave better results for IM AGBs undergoing HBB, would this imply that the calculations using set B for LM stars are more realistic? In other words, the best fits for both LM and IM stars should use the same rate set if both types of stars contributed these grains. Otherwise it would imply that IM stars undergoing HBB are not likely sources of Group 2 grains. What does the study show about the two different reaction rate sets? What are the overall implications of the conclusions?

Author Response

(The authors gave the same response as above.)

Reviewer 3 Report

Dear authors, 

I have read the manuscript with great interest. The paper is very interesting to me, offering other alternatives/possibilities (although in my opinion, still
subject to the validity/reality of the models presented and at odds with present astronomical observations) to explain the origin of the enigmatic group II grains. I am giving some first comments and suggestions, which I hope could encourage the authors to improve their work. I would like to see a revised version addressing my comments/suggestions, as enumerated below. 

***Title:

Perhaps, the title could be: Type II oxide grains: how much massive are AGB stars they come from?

***General Comments

I think that, along the text, it should be more clear that the paper results and
interpretation are based on some kind of models (FRUITY? other?) and subject to its validity/reality. Also, those models should be, at least briefly, better explained/described. 

I miss a mention to the main differences when comparing with other nucleosynthesis codes in the literature (e.g., Monash, ATON, NuGrid, ...). I think that this should be critically evaluated in the paper (at least briefly). For example, what is the effect of the HBB strength in the model predictions presented here? HBB is very weak in FRUITY models but much stronger in Monash or ATON models.

Also, I think that the paper would be more fair if some criticisms are given for all the possibilities (all models depending on progenitor mass; low-mass vs high-mass) presented here. A reading gives the impression that the low-mass models are perfect and very good with no oversimplified assumptions, while the contrary is the case for the high-mass models. 

Finally, I miss a fair comparison and discussion with astronomical observations of AGB stars. As far as I know, there are no precise and uncertain measurements of O isotopic ratios in truly massive HBB AGB stars in the refereed literature (such stars are very difficult to model, especially in the near-IR where the CNO isotopic ratios can be measured). Recent studies of different kind of O-rich variables do not generally found stars (with only a few exceptions, 1-2 stars) with the characteristic O isotopic ratios of type II grains (Hinkle et al. 2016, ApJ, 825, 38; Lebzelter et al. 2019, ApJ, 886, 117). The observed stars are interpreted as low-mass AGB stars. I mean that such observed samples of AGBs (indendependly of their true progenitor mass, although they seem to be of low-mass) do not occupy the O isotopic composition of type II grains. Here the authors, put forward the possibility that low-mass AGB stars can explain the type II grains, but this seems to be at odds with present astronomical observational evidence. These facts could be discussed in the paper. 

See also another comments/suggestions below, as they appear in the text.

**Abstract

I think that the preferred answer of the authors could be given in the abstract.

**Section 1

In the text: ....provided by IM-AGBs with HBB [27]... 

I think that the reference [27] here should be Lugaro et al. (2017, NatAs, your final reference [58]). Is not? If yes, then the right reference and the references order should be corrected.

Here, I also miss a better explanation of the updates of this work with respect to those of [27] (Lugaro et al.) and [28] (Palmerini et al.).

**Section 3

**Figure 3.

How the IM-AGBs predictions change with k in this figure? I see those variations for the lower mass models but they seem to be lacking (and explained) for the higher mass ones.

In the text: At the moment in which this note is prepared, the only LM-AGB
nucleosynthesis model able to simultaneously reproduce the 17 O/ 16 O, the 18O/16O and the 26Al/27Al ratios measured in group 2 oxide grains is the one suggested by Busso et al. [43] and formalized by Nucci and Busso [44].

I guess this only refers to low-mass AGB stars. Lugaro et al. 2017 specifically
say that "..the high 26Al/27Al ratios up to ~0.1 typical of Group II grains are
also consistent with HBB (Figure 4b), although an accurate analysis is currently hampered by the uncertainties in the 25Mg and 26Al proton-capture
rates (Iliadis et al. 2010, Straniero et al. 2010)." So, are these uncertainties not affecting to the low-mass AGB models? Please clarify and/or correct.

In the text: ...are reported in Figure 3, 4 and 5 for..

Please explain Fig. 5 before mentioning it in the text.

**Section 4

What are the k values that could work for your massive HBB AGB models (e.g., in your Fig. 3 about the O isotopic ratios)?

As mentioned above, many criticisms are given here, while the same was not previously done with the low-mass AGB models presented before. I am aware that the agreement (between different models/codes) when modeling low-mass AGB stars is better but, if I understood well, the only low-mass AGB models able to reproduce the O and Al isotopic ratios are those presented here. So, maybe the advantages and caveats of these models could be mentioned/explained. 

In the text: ...of a star should be solved simultaneously [52].

This seems to be a paper in preparation. Any update? or more information?

In the text: We make use of the FUNS code [3,53-57], adopting...

What is FUNS? Is this the name of the code presented here? Is this the same of FRUITY? This is the first time that a code name is given along the text.

What are the mass loss values and mass loss prescription used in the modeling of HBB AGBs? (And for the low-mass AGBs?)

What happen in the your models for more realistic (or higher) mass-loss values? Are the IM-AGB models also including magnetic induced mixing? Likely these issues need to be discussed and clarified.

In the text: Interestingly enough, [58] suggested the possibility...

I guess this should be reference [27] given before (see above).

**Conclusions

They probably should be edited at the end, once the comments above have been addressed.

Author Response

(The authors gave the same response as above.)

Reviewer 4 Report

This is a difficult topic, given the level of uncertainty in data and models.   This should be commented on and the authors should present their work as the testing of an hypothesis.  This should be stated in the Abstract as should be your results.

Yuo should comment on the validity of using solar abundances in your models

whose origin is difficult to be addressed - is difficult to address

What are the the group 1 grains - are these discussed in the paper as promised?

SDU?  Please define

TDU? please define

is mandatorily induced - what does this mean?

We show only values resulting by using rate recommended, being these former the input employed : Please correct the language in italics

trying to figure out if - replace with "in order to determine if"

Author Response

(The authors gave the same response as above.)

Round 2

Reviewer 3 Report

Dear authors, 

I acknowledge very much the author's efforts. The paper has been improved with respect to the previous version, although I would have preferred an author's reply point by point (this would save time and it would be easier to evaluate the corrections/changes along the revised version).

Indeed, I think we are very close but I am still not completely happy with the
revised version and I still have a few main general comments (some already in
the previous report) and a few minor ones (most related with the new text and
data that the authors have included in the revised version of the paper) that
should be addressed before the paper could be accepted for publication.

****Main comments:

To me this paper is showing that FUNS models can reproduce the bulk of group II
grains with very low-mass AGB models (1.2 and 1.5 Msun) with deep mixing. The
results are not completely indepedent of nuclear physics (with the set A, Trojan
Horse, of reaction rates been favored), being dependent on the k parameter. FUNS
models of high-mass HBB AGBs cannot reproduce the bulk of group II grains (only
those with 4.5 and 5 Msun with the set B, LUNA, of reaction rates could
reproduce some of the most extreme grains -- curiously those that do not have a
counterpart among the available observations of low-mass AGB stars). The
abstract and the conclusions need some final adjustments to reflect this (see my
specific suggestions below). At least, it should be clear that everything is
according to the FUNS models. To me, this is the fair story of this paper and it
is a good one! However, it is not completely clear in the revised version of the
paper. 

The comparison of the presolar grain data with the astronomical observations show that the low-mass AGB data do not reach the extreme O18 depletions observed in the presolar grains. I have provided specific corrections/suggestions below (minor comments) in order to reflect this.

The FUNS models somehow favor the Set A of reaction rates (especially for the O
isotopic ratios). However, I do not see any mention to why one should use the
Set A (Trojan Horse Method) instead of the Set B (LUNA) that cames from direct
measurements. Convincing arguments in favor of Set A could be given or this
possible caveat could be clearly mentioned.

****Minor comments/suggestions:

*Title: Group II oxide grains: how massive are AGB stars and where do they
come from? (Referee: I think that this should be the right title)

*Abstract

-"...provide the most valuable constraints..." --> provide valuable constraints

-"Nuclear physics might indicate the most likely candidate, which could be low
mass stars, because their fit to presolar grains seems to be more accurate and
less sensitive to the nuclear inputs adopted." --> Our FUNS models of low-mass AGB stars with a bottom-up deep mixing - being less sensitive than their intermediate-mass counterparts to the nuclear physics inputs - are shown to be possible progenitors of Group II grains.  

*Introduction:

-"...and silicon grains found..." --> silicate grains ? Referee: also, do you mean amorphous or crystalline silicates or both?. Please specify.

-"However, traces of Mg and Al have been so far recorded only
in presolar oxide grains. As a consequence, in this note we focus on oxide grains only, and in particular those belonging to group 2,..." --> Referee: Why those traces have not been recorded in the silicate grains? It is just a technical problem? Not tried? Please clarify. Also, how a focus only on oxide grains could bias your sample and interpretation? This should be explained. For example, it seems that there is a lack of Group II grains (your new Figure 2) among the presolar silicate grains, why? I wonder if this could indicate a real effect in the sense that oxide and silicate grains are preferentially formed for different progenitor masses. According to their references (e.g. 7,8 among others) Al2O3 is theoretially expected to be practically not formed in very low-mass (<1.5 Msun) AGBs but being strongly formed in high-mass HBB AGBs. I would like the authors to comment and/or clarify this.

-"...objects with initial mass <= 2 Msun affected by CBP.." --> Referee: should the mass be 1.5 Msun instead of 2? You later say in the text 1.5 Msun.

*Section 2

-"...in massive HBB AGB.." --> in dusty and truly massive (> 4 Msun) HBB AGB

-"Recent studies of Mira, SRa- and Lb-type variable AGB stars [34,35].." --> Recent studies of different types of variable (Mira-, SRa-, SRb- and Lb-type) AGB stars [34,35]

-"..for a relatively large (~40).." --> Referee: this number should be revised. Only in the first paper ([34]) there are 46 stars.

-"...no stars affected by HBB were observed." --> no stars affected by HBB were observed, in agreement with their estimated low progenitor masses.

-"...not large enough to confirm (or reject) that their oxygen isotope mix is
compatible with that of group 2 oxide grains." --> not good enough to
unambiguously confirm (or reject) that their oxygen isotope mix is compatible
with that of group 2 oxide grains with O18/O16 > 10-4. On the other hand, the
available astronomical data suggest that LM AGBs do not explain the most extreme
and O18-depleted group II grains with O18/O16 < 10-4.

-"..oxide isotopic ratios.." --> oxygen isotopic ratios

-"...according to the technique (curve of growth or spectrum synthesis) employed
in data reduction." --> depending on the technique (curve of growth or spectrum
synthesis) employed in chemical abundance analysis. 

*Section 4

-"Although these values are sufficient to reproduce a large part of the
26Al/27Al spread shown by group 2 oxide grains, the models reach the highest
26Al abundances when they are already depleted in 18 O, and thus unable to match
the grains." --> Referee: I am confused here. Do you refer to your FUNS massive HBB star models? or those from Lugaro et al.? In this regard, the behaviour of the Lugaro et al. HBB star models is identical to the FUNS models presented here? Please clarify.

*Section 5

-"..these massive (but not much) objects." --> the more massive AGB stars.

-"...the theoretical 18O/16O ratios are extremely low (< 10-7)." --> the final theoretical 18O/16O ratios are extremely low (< 10-7).

*Section 6

-"Low mass AGB stars with a bottom-up deep mixing..." --> Low-mass AGB FUNS
models with a bottom-up deep mixing 

-"...are likely candidates..." --> are possible candidates

-"...group 2, independently from the adopted nuclear physics inputs." --> group
2. 

-"...intermediate mass AGB models.." --> FUNS intermediate-mass AGB models

Author Response

Dear referee,

once again many thanks for your careful review. We tried to address all your suggestions and the replay point by point follows.

Before entering in detail in describing the corrections we would like to give a general answer to your main comments.

Our IM stars (with HBB) are modelled by the FUNS code, but this is not completely true for our LM AGB models with magnetic mixing. Indeed, the LM stellar structure are computed by FUNS, but the extra-mixing (and its effects) is calculated by running on these structures the MAGIC post process code, as it is in Palmerini et al 2017. Since it resulted unclear, we have now explained it explicitly at the end of session 1.

Second, we conclude that LMS AGBs are most likely progenitors of group 2 oxide grains because our LMS models fit the 17O/16O, 18O/16O and 26Al/27Al values recorded in the grains altogether, while IMS one fail at least in accounting 18O/16O and 26Al/27Al at the same time. In this framework, one achieves a better agreement between grain abundances and models, if these are run by using nuclear data of the Set A instead of ones of the Set B. We agree with you that is more appropriate to emphasize the role of the models before that of nuclear physics input and in doing that we correct the sentence “However, the key element to identify the masses of stellar progenitors could come from nuclear physics” writing in its place “However, a key role in identifying the masses of stellar progenitors is also played by nuclear physics.”.

****Minor comments/suggestions:

*Title: Group II oxide grains: how massive are AGB stars and where do they
come from? (Referee: I think that this should be the right title)

REPLAY:

We changed the title in “group ii oxide grains: how massive are their AGB star progenitors?” because we would like to avoid any possible misunderstanding: we are  dealing with the mass of the stars that can be progenitors of group 2 grains and not of the whole mass range of the stars undergoing the AGB phase.

*Abstract

-"...provide the most valuable constraints..." --> provide valuable constraints 

REPLAY: done

-"Nuclear physics might indicate the most likely candidate, which could be low
mass stars, because their fit to presolar grains seems to be more accurate and
less sensitive to the nuclear inputs adopted." --> Our FUNS models of low-mass AGB stars with a bottom-up deep mixing - being less sensitive than their intermediate-mass counterparts to the nuclear physics inputs - are shown to be possible progenitors of Group II grains.  

REPLAY:

We rewrote the sentence as follows: “our models of low-mass AGB stars with a bottom-up deep mixing are shown to be likely progenitors of group 2 grains, reproducing together the 17o/16o, 18o/16o and 26al/27al values found in those grains and being less sensitive than our intermediate-mass models with HBB to the nuclear physics input.”  Indeed, we are suggesting that LM AGBs (with bottom-up extra mixing at play) are the most reliable progenitor of group 2 grains mainly because these models provide a better fit to the oxygen and aluminum isotopic ratios, fitting all the 3 isotopic ratios at the same time (see fig 4, 5 and 6) while IMS models with HBB fail in accounting for the 3 isotopic ratios at the same time, in particular the 18o/16o and the 26al/27al. The minor dependence to nuclear inputs is a second argument in favor of our thesis. Moreover it is not exact to deal with “FUNS models of low-mass AGB stars with a bottom-up deep mixing“ because the magnetic mixing has been modelled by  the MAGIC post-process code run on stellar structure computed by FUNS (we explained that at the end of section 1).

*Introduction:

-"...and silicon grains found..." --> silicate grains ? Referee: also, do you mean amorphous or crystalline silicates or both?. Please specify.

REPLAY: both of them….we wrote it

-"However, traces of Mg and Al have been so far recorded only
in presolar oxide grains. As a consequence, in this note we focus on oxide grains only, and in particular those belonging to group 2,..." --> Referee: Why those traces have not been recorded in the silicate grains? It is just a technical problem? Not tried? Please clarify also, how a focus only on oxide grains could bias your sample and interpretation? This should be explained. For example, it seems that there is a lack of Group II grains (your new Figure 2) among the presolar silicate grains, why? I wonder if this could indicate a real effect in the sense that oxide and silicate grains are preferentially formed for different progenitor masses. According to their references (e.g. 7,8 among others) Al2O3 is theoretically expected to be practically not formed in very low-mass (<1.5 Msun) AGBs but being strongly formed in high-mass HBB AGBs. I would like the authors to comment and/or clarify this.

REPLAY:

We are not expert in presolar grain analysis thus we prefer to not discuss among technical issues that could hamper the searching Al or Mg traces in silicate grains. Therefore, we try to address the referee requirement writing:  “Over the years, also silicate grains have been collected  in the presolar grain database of the Washington University of Saint Luis  [14]. As it is shown in Figure 2A, they have the same oxygen isotopic mix of the oxide grains and they can be classified in four groups as well. Although, the distribution of the silicate grains in the four groups is not the same of the oxide ones. In particular, group 4 silicates are overabundant compared to those of groups 1 and 2 (probably because of the injection of materials from a nearby supernova into the solar nebula [20]); while group 2 silicates are rarer than oxides. Such a lack could be accounted for by the fact that silicates are expected to form mainly in IM AGBs affected by HBB [7,8].

According to the WUSTL Database, traces of Mg and Al have been so far recorded only in oxide  grains and not in silicate ones. As we will discuss in the next sections (and as illustrated by Figures 5 and 6), the isotopic ratio of aluminum plays a fundamental role in investigating  the mass of the stellar progenitors of the grains. Therefore, in this note we focus on oxide grains only, and in particular those belonging to group 2, which have retained to have AGB origins and  show 17O/16O ratios larger than the solar value and 18O/16O<0.001.”

-"...objects with initial mass <= 2 Msun affected by CBP.." --> Referee: should the mass be 1.5 Msun instead of 2? You later say in the text 1.5 Msun.

REPLAY: Done

*Section 2

-"...in massive HBB AGB.." --> in dusty and truly massive (> 4 Msun) HBB AGB

REPLAY: almost done, we wrote “in dusty and IM (> 4 Mo) HBB AGB” indeed stars with mass from 4 to 6.5Mo are just IMS not massive stars.

-"Recent studies of Mira, SRa- and Lb-type variable AGB stars [34,35].." --> Recent studies of different types of variable (Mira-, SRa-, SRb- and Lb-type) AGB stars [34,35]

REPLAY: Done

-"..for a relatively large (~40).." --> Referee: this number should be revised. Only in the first paper ([34]) there are 46 stars.

REPLAY: the stars useful for the comparisons with grains; namely the O-rich ones for which the data for both the oxygen isotopic ratio are available, are 77 as 77 are the stars reported in figure 2b

-"...no stars affected by HBB were observed." --> no stars affected by HBB were observed, in agreement with their estimated low progenitor masses.  

REPLAY: we are sorry we are not able to address this point indeed the authors of papers  [34] and [35] discard the hypothesis that stars in their sample are affected by HBB because the measured values of the 12C/13C isotopic ratio are too large to  be compatible with HBB models. Thus, they conclude that the stars being O-rich and showing no evidence of HBB in their C-isotopic ratios are lMS.

-"...not large enough to confirm (or reject) that their oxygen isotope mix is
compatible with that of group 2 oxide grains." --> not good enough to
unambiguously confirm (or reject) that their oxygen isotope mix is compatible
with that of group 2 oxide grains with O18/O16 > 10-4. On the other hand, the
available astronomical data suggest that LM AGBs do not explain the most extreme
and O18-depleted group II grains with O18/O16 < 10-4.

REPLAY: we wrote something similar adding a comment “Therefore, although the observations of oxygen isotopic ratios in LM AGBs are valuable data, their precision is not good enough to unambiguously confirm (or reject) that their oxygen isotope mix is compatible with that of group 2 oxide grains. In particular, the available astronomical data seem to suggest that LM AGBs do not explain the most extreme and 18O-depleted group 2 grains with 18O/16O<10^-4. However, one might speculate that this value is the lower limit within which it is currently possible to measure the 18O/16O ratio in the spectrum of AGB stars.”

-"..oxide isotopic ratios.." --> oxygen isotopic ratios  

REPLAY:done

-"...according to the technique (curve of growth or spectrum synthesis) employed
in data reduction." --> depending on the technique (curve of growth or spectrum
synthesis) employed in chemical abundance analysis. 

REPLAY:done

*Section 4

-"Although these values are sufficient to reproduce a large part of the
26Al/27Al spread shown by group 2 oxide grains, the models reach the highest
26Al abundances when they are already depleted in 18 O, and thus unable to match
the grains." --> Referee: I am confused here. Do you refer to your FUNS massive HBB star models? or those from Lugaro et al.? In this regard, the behaviour of the Lugaro et al. HBB star models is identical to the FUNS models presented here? Please clarify.

REPLAY: both our models and the ones in Lugaro et al have the same problem. We wrote it.

*Section 5

-"..these massive (but not much) objects." --> the more massive AGB stars. S

REPLAY: done.

-"...the theoretical 18O/16O ratios are extremely low (< 10-7)." --> the final theoretical 18O/16O ratios are extremely low (< 10-7). 

REPLAY: done.

*Section 6

-"Low mass AGB stars with a bottom-up deep mixing..." --> Low-mass AGB FUNS
models with a bottom-up deep mixing

REPLAY: As we have explained before this are not  FUNS models.  Moreover, as it is discussed  in detail by  [31], we are confident that not only our LM AGB models with a magnetic mixing, but whatever LMS AGB model with an advective bottom-up deep mixing might provide a good fit to group 2 oxide grains.

-"...are likely candidates..." --> are possible candidates

REPLAY: done.

-"...group 2, independently from the adopted nuclear physics inputs." --> group
2. 

REPLAY: done.

-"...intermediate mass AGB models.." --> FUNS intermediate-mass AGB models

REPLAY: done.